# WebArena: A Realistic Web Environment for Building Autonomous Agents

**Shuyan Zhou**[*]   **Frank F. Xu**[*]
**Hao Zhu** [†]   **Xuhui Zhou**[†]   **Robert Lo**[†]   **Abishek Sridhar**[†]   **Xianyi Cheng**
**Tianyue Ou**   **Yonatan Bisk**   **Daniel Fried**   **Uri Alon**   **Graham Neubig**

Carnegie Mellon University
`{shuyanzh, fangzhex, gneubig}@cs.cmu.edu`

## Abstract

With advances in generative AI, there is now potential for autonomous agents to manage daily tasks via natural language commands. However, current agents are primarily created and tested in simplified synthetic environments, leading to a disconnect with real-world scenarios. In this paper, we build an environment for language-guided agents that is *highly realistic* and *reproducible*. Specifically, we focus on agents that perform tasks on the web, and create an environment with fully functional websites from four common domains: e-commerce, social forum discussions, collaborative software development, and content management. Our environment is enriched with tools (*e.g.,* a map) and external knowledge bases (*e.g.,* user manuals) to encourage human-like task-solving. Building upon our environment, we release a set of benchmark tasks focusing on evaluating the *functional correctness* of task completions. The tasks in our benchmark are diverse, long-horizon, and designed to emulate tasks that humans routinely perform on the internet. We experiment with several baseline agents, integrating recent techniques such as reasoning before acting. The results demonstrate that solving complex tasks is challenging: our best GPT-4-based agent only achieves an end-to-end task success rate of 14.41%, significantly lower than the human performance of 78.24%. These results highlight the need for further development of robust agents, that current state-of-the-art large language models are far from perfect performance in these real-life tasks, and that `WebArena` can be used to measure such progress.

Our code, data, environment reproduction resources, and video demonstrations are publicly available at `https://webarena.dev/`.

## 1 Introduction

Autonomous agents that perform everyday tasks via human natural language commands could significantly augment human capabilities, improve efficiency, and increase accessibility. Nonetheless, to fully leverage the power of autonomous agents, it is crucial to understand their behavior within an environment that is both *authentic* and *reproducible*. This will allow measurement of the ability of agents on tasks that human users care about in a fair and consistent manner.

Current environments for evaluate agents tend to *over-simplify* real-world situations. As a result, the functionality of many environments is a limited version of their real-world counterparts, leading to a lack of task diversity (Shi et al., 2017; Anderson et al., 2018; Gordon et al., 2018; Misra et al., 2016; Shridhar et al., 2020; 2021; Yao et al., 2022a). In addition, these simplifications often lower the complexity of tasks as compared to their execution in the real world (Puig et al., 2018; Shridhar et al., 2020; Yao et al., 2022a). Finally, some environments are presented as a static resource (Shi et al., 2017; Deng et al., 2023) where agents are confined to accessing only those states that were previously cached during data collection, thus limiting the breadth and diversity of exploration. For evaluation, many environments focus on comparing the textual *surface form* of the predicted

---

[*]Lead contributors.
[†]Equal contribution.

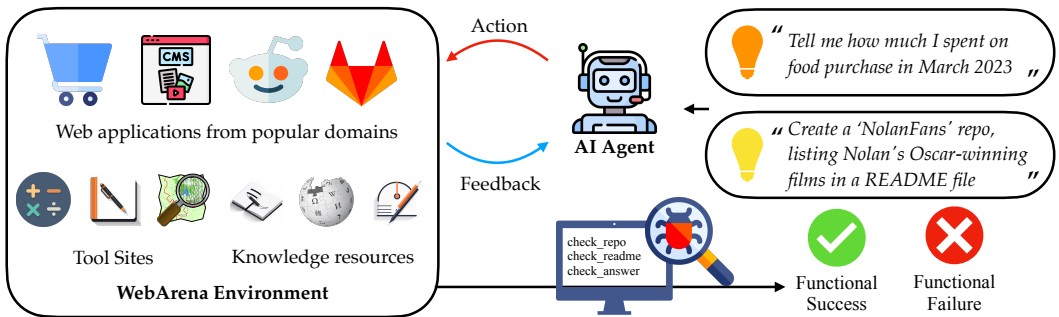

Figure 1: `WebArena` is a standalone, self-hostable web environment for building autonomous agents. `WebArena` creates websites from four popular categories with functionality and data mimicking their real-world equivalents. To emulate human problem-solving, `WebArena` also embeds tools and knowledge resources as independent websites. `WebArena` introduces a benchmark on interpreting *high-level realistic* natural language command to concrete web-based interactions. We provide validators to programmatically validate the functional correctness of each task.

action sequences with reference action sequences, disregarding the *functional correctness* of the executions and possible alternative solutions (Puig et al., 2018; Jernite et al., 2019; Xu et al., 2021; Li et al., 2020; Deng et al., 2023). These limitations often result in a discrepancy between simulated environments and the real world, and can potentially impact the generalizability of AI agents to successfully understand, adapt, and operate within complex real-world situations.

We introduce `WebArena`, a *realistic* and *reproducible* web environment designed to facilitate the development of autonomous agents capable of executing tasks (§2). An overview of `WebArena` is in Figure 1. Our environment comprises four fully operational, self-hosted web applications, each representing a distinct domain prevalent on the internet: online shopping, discussion forums, collaborative development, and business content management. Furthermore, `WebArena` incorporates several utility tools, such as map, calculator, and scratchpad, to best support possible human-like task executions. Lastly, `WebArena` is complemented by an extensive collection of documentation and knowledge bases that vary from general resources like English Wikipedia to more domain-specific references, such as manuals for using the integrated development tool (Fan et al., 2022). The content populating these websites is extracted from their real-world counterparts, preserving the authenticity of the content served on each platform. We deliver the hosting services using Docker containers with `gym`-APIs (Brockman et al., 2016), ensuring both the usability and the reproducibility of `WebArena`.

Along with `WebArena`, we release a ready-to-use benchmark with 812 long-horizon web-based tasks (§3). Each task is described as a high-level natural language intent, emulating the abstract language usage patterns typically employed by humans (Bisk et al., 2019). Two example intents are shown in the upper left of Figure 1. We focus on evaluating the *functional correctness* of these tasks, *i.e.,* does the result of the execution actually achieve the desired goal (§3.2). For instance, to evaluate the example in Figure 2, our evaluation method verifies the concrete contents in the designated repository. This evaluation is not only more reliable (Zhong et al., 2017; Chen et al., 2021; Wang et al., 2022) than comparing the textual surface-form action sequences (Puig et al., 2018; Deng et al., 2023) but also accommodate a range of potential valid paths to achieve the same goal, which is a ubiquitous phenomenon in sufficiently complex tasks.

We use this benchmark to evaluate several agents that can follow NL command and perform web-based tasks (§4). These agents are implemented in a few-shot in-context learning fashion with powerful large language models (LLMs) such as GPT-4 and PALM-2. Experiment results show that the best GPT-4 agent performance is somewhat limited, with an end-to-end task success rate of only 14.41%, while the human performance is 78.24%. We hypothesize that the limited performance of current LLMs stems from a lack of crucial capabilities such as active exploration and failure recovery to successfully perform complex tasks (§5.1). These outcomes underscore the necessity for further development towards robust and effective agents (LeCun, 2022) in `WebArena`.

## 2    WEBARENA: WEBSITES AS AN ENVIRONMENT FOR AUTONOMOUS AGENTS

Our goal is to create a *realistic* and *reproducible* web environment. We achieve reproducibility by making the environment standalone, without relying on live websites. This circumvents technical

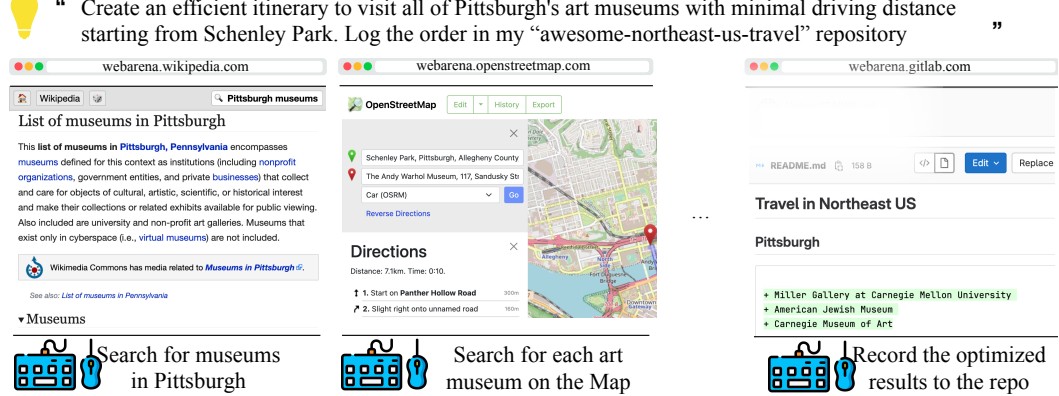

Figure 2: A high-level task that can be fully executed in `WebArena`. Success requires sophisticated, long-term planning and reasoning. To accomplish the goal (top), an agent needs to (1) find Pittsburgh art museums on Wikipedia, (2) identify their locations on a map (while optimizing the itinerary), and (3) update the README file in the appropriate repository with the planned route.

challenges such as bots being subject to CAPTCHAs, unpredictable content modifications, and configuration changes, which obstruct a fair comparison across different systems over time. We achieve realism by using open-source libraries that underlie many in-use sites from several popular categories and importing data to our environment from their real-world counterparts.

## 2.1 Controlling Agents through High-level Natural Language

The `WebArena` environment is denoted as $\mathcal{E} = \langle \mathcal{S}, \mathcal{A}, \mathcal{O}, \mathcal{T} \rangle$ with state space $\mathcal{S}$, action space $\mathcal{A}$ (§2.4) and observation space $\mathcal{O}$ (§2.3). The transition function $\mathcal{T} : \mathcal{S} \times \mathcal{A} \longrightarrow \mathcal{S}$ is deterministic, and it is defined by the underlying implementation of each website in the environment. Given a task described as a natural language intent $\mathbf{i}$, an agent issues an action $a_t \in \mathcal{A}$ based on intent $\mathbf{i}$, the current observation $o_t \in \mathcal{O}$, the action history $\mathbf{a}_1^{t-1}$ and the observation history $\mathbf{o}_1^{t-1}$. Consequently, the action results in a new state $s_{t+1} \in \mathcal{S}$ and its corresponding observation $o_{t+1} \in \mathcal{O}$. We propose a reward function $r(\mathbf{a}_1^T, \mathbf{s}_1^T)$ to measure the success of a task execution, where $\mathbf{a}_1^T$ represents the sequence of actions from start to the end time step $T$, and $\mathbf{s}_1^T$ denotes all intermediate states. This reward function assesses if state transitions align with the expectations of the intents. For example, with an intent to place an order, it verifies whether an order has been placed. Additionally, it evaluates the accuracy of the agent's actions, such as checking the correctness of the predicted answer.

## 2.2 Website Selection

To decide which categories of websites to use, we first analyzed approximately 200 examples from the authors' actual web browser histories. Each author delved into their browsing histories, summarizing the goal of particular segments of their browser session. Based on this, we classified the visited websites into abstract categories. We then identified the four most salient categories and implemented one instance per category based on this analysis: (1) E-commerce platforms supporting online shopping activities (*e.g.,* Amazon, eBay), (2) social forum platforms for opinion exchanges (*e.g.,* Reddit, StackExchange), (3) collaborative development platforms for software development (*e.g.,* GitLab), and (4) content management systems (CMS) that manage the creation and revision of the digital content (*e.g.,* online store management).

In addition to these platforms, we selected three utility-style tools that are frequently used in web-based tasks: (1) a map for navigation and searching for information about points of interest (POIs) such as institutions or locations (2) a calculator, and (3) a scratchpad for taking notes. As information-seeking and knowledge acquisition are critical in web-based tasks, we also incorporated various knowledge resources into `WebArena`. These resources range from general information hubs, such as the English Wikipedia, to more specialized knowledge bases, such as the website user manuals.

**Implementation** We leveraged open-source libraries relevant to each category to build our own versions of an E-commerce website (OneStopShop), GitLab, Reddit, an online store content management system (CMS), a map, and an English Wikipedia. Then we imported sampled data from their

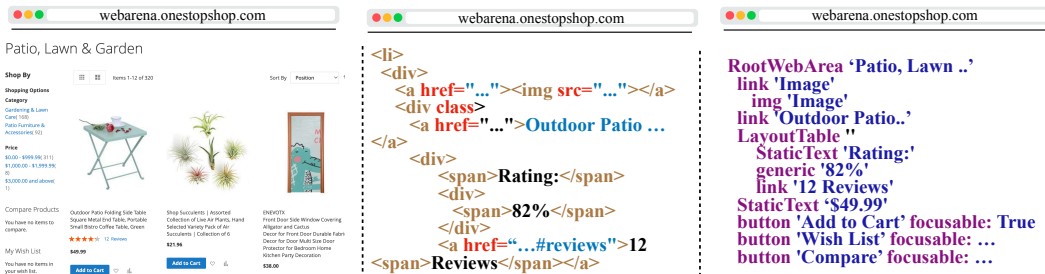

Figure 3: We design the observation to be the URL and the content of a web page, with options to represent the content as a screenshot (left), HTML DOM tree (middle), and accessibility tree (right). The content of the middle and right figures are trimmed to save space.

real-world counterparts. As an example, our version of GitLab was developed based on the actual GitLab project.[1] We carefully emulated the features of a typical code repository by including both popular projects with many issues and pull requests and smaller, personal projects. Details of all websites in `WebArena` can be found in Appendix A.1. We deliver the environment as dockers and provide scripts to reset the environment to a deterministic initial state (See Appendix A.2).

## 2.3 OBSERVATION SPACE

We design the observation space to roughly mimic the web browser experience: a web page URL, the opened tabs , and the web page content of the focused tab. `WebArena` is the first web environment to consider multi-tab web-based tasks to promote tool usage, direct comparisons and references across tabs, and other functionalities. The multi-tab functionality offers a more authentic replication of human web browsing habits compared to maintaining everything in a single tab. We provide flexible configuration to render the page content in many modes: (see Figure 3 for an example): (1) the raw web page HTML, composed of a Document Object Model (DOM) tree, as commonly used in past work (Shi et al., 2017; Deng et al., 2023; Li et al., 2020); (2) a screenshot, a pixel-based representation that represents the current web page as an RGB array and (3) the accessibility tree of the web page.[2] The accessibility tree is a subset of the DOM tree with elements that are *relevant* and *useful* for displaying the contents of a web page. Every element is represented as its role (*e.g.,* a link), its text content, and its properties (*e.g.,* whether it is focusable). Accessibility trees largely retain the *structured* information of a web page while being more compact than the DOM representation.

We provide an option to limit the content to the contents within a viewport for all modes. This ensures that the observation can be input into a text-based model with limited context length or an image-based model with image size or resolution requirements.

## 2.4 ACTION SPACE

Following previous work on navigation and operation in web and embodied environments (Shi et al., 2017; Liu et al., 2018), we design a compound action space that emulates the keyboard and mouse operations available on web pages. Figure 4 lists all the available actions categorized into three distinct groups. The first group includes element operations such as clicking, hovering, typing, and key combination pressing. The second comprises tab-related actions such as opening, closing, and switching between tabs. The third category consists of URL navigation actions, such as visiting a specific URL or navigating forward and backward in the browsing history.

Building on these actions, `WebArena` provides agents with the flexibility to refer to elements for operation in different ways. An element can be selected by its on-screen coordinates, $(x, y)$, or by a unique element ID that is prepended to each element. This ID is generated when traversing the Document Object Model (DOM) or accessibility tree. With element IDs, the element selection is transformed into an $n$-way classification problem, thereby eliminating any disambiguation efforts required from the agent or the underlying implementation. For example, issuing the action `click [1582]` clicks the button given the observation of `[1582] Add to Cart`. This flexible element selection allows `WebArena` to support agents designed in various ways (*e.g.,* accepting input from different modalities) without compromising fair comparison metrics such as step count.

---

[1] `https://gitlab.com/gitlab-org/gitlab`
[2] `https://developer.mozilla.org/en-US/docs/Glossary/Accessibility_tree`

| Action Type | Description |
|---|---|
| `noop` | Do nothing |
| `click(elem)` | Click at an element |
| `hover(elem)` | Hover on an element |
| `type(elem, text)` | Type to an element |
| `press(key_comb)` | Press a key comb |
| `scroll(dir)` | Scroll up and down |
| `tab_focus(index)` | focus on $i$-th tab |
| `new_tab` | Open a new tab |
| `tab_close` | Close current tab |
| `go_back` | Visit the last URL |
| `go_forward` | Undo `go_back` |
| `goto(URL)` | Go to URL |

Figure 4: Action Space of `WebArena`

| Category | Example |
|---|---|
| Information Seeking | When was the last time I bought shampoo |
| | Compare walking and driving time from AMC Waterfront to Randyland |
| Site Navigation | Checkout merge requests assigned to me |
| | Show me the ergonomic chair with the best rating |
| Content & Config | Post to ask "whether I need a car in NYC" |
| | Delete the reviews from the scammer Yoke |

Figure 5: Example intents from three categories.

**User Role Simulation**   Users of the same website often have disparate experiences due to their distinct *roles*, *permissions*, and *interaction histories*. We emulate this scenario by generating unique user profiles on each platform. The details can be found in Appendix A.3.

## 3    BENCHMARK SUITE OF WEB-BASED TASKS

We provide a benchmark with 812 test examples on grounding high-level natural language instructions to interactions in `WebArena`. Each example has a metric to evaluate the functional correctness of the task execution. In this section, we first formally define the task of controlling an autonomous agent through natural language. Then we introduce the annotation process of our benchmark.

### 3.1    INTENT COLLECTION

We focus on curating *realistic* intents to carry out *complex* and *creative* tasks within `WebArena`. To start with, our annotators were guided to spend a few minutes exploring the websites to familiarize themselves with the websites' content and functionalities. As most of our websites are virtually identical to their open-web counterparts, despite having sampled data, most annotators can quickly comprehend the websites.

Next, we instructed the annotators to formulate intents based on the following criteria:

(1) The intent should be *abstract* and *high-level*, implying that the task cannot be fulfilled with merely one or two actions. As an example, instead of "*click the `science` subreddit*", we encouraged annotators to come up with something more complex like "*post a greeting message on `science` subreddit*", which involves performing multiple actions.
(2) The intent should be *creative*. Common tasks such as account creation can be easily thought of. We encouraged the annotators to add constraints (*e.g.,* "*create a Reddit account **identical to my GitLab one**"*) to make the intents more unique.
(3) The intent should be formulated as a *template* by making replaceable elements as variables. The annotators were also responsible for developing several instantiations for each variable. For example, the intent "*create a Reddit account identical to my GitLab one*" can be converted into "*create a {{site1}} account identical to my {{site2}} one*", with an instantiation like "*{site1: Reddit, site2: GitLab}*" and another like "*{site1: GitLab, site2: OneStopShopping}*". Notably, tasks derived from the same template can have distinct execution traces. The similarity resides primarily in the high-level semantics rather than the specific implementation.

We also provided a prompt for the annotators to use with ChatGPT[3] for inspiration, that contains an overview of each website and instructs the model to describe potential tasks to be performed on these sites. Furthermore, we offered a curated list of examples for annotators to reference.

**Intent Analysis**   In total, we curated 241 templates and 812 instantiated intents. On average, each template is instantiated to 3.3 examples. The intent distribution is shown in Figure 6. Furthermore, we classify the intents into three primary categories with examples shown in Figure 5:

---

[3]`https://chat.openai.com/`

(1) **Information-seeking** tasks expect a textual response. Importantly, these tasks in `WebArena` often require navigation across multiple pages or focus on *user-centric* content. This makes them distinct from open-domain question-answering (Yang et al., 2018; Kwiatkowski et al., 2019), which focuses on querying general knowledge with a simple retrieval step. For instance, to answer "*When was the last time I bought the shampoo*", an agent traverses the user's purchase history, checking order details to identify the most recent shampoo purchase.

(2) **Site navigation**: This category is composed of tasks that require navigating through web pages using a variety of interactive elements such as search functions and links. The objective is often to locate specific information or navigate to a particular section of a site.

(3) **Content and configuration operation**: This category encapsulates tasks that require operating in the web environment to create, revise, or configure content or settings. This includes adjusting settings, managing accounts, performing online transactions, generating new web content, and modifying existing content. Examples range from updating a social media status or README file to conducting online purchases and configuring privacy settings.

## 3.2 EVALUATION ANNOTATION

**Evaluating Information Seeking Tasks**   To measure the correctness of information-seeking tasks where a textual answer is expected, we provide the annotated answer $a^*$ for each intent. The $a^*$ is further compared with the predicted answer $\hat{a}$ with one of the following scoring functions $r_{\text{info}}(\hat{a}, a^*)$.

First, we define `exact_match` where only $\hat{a}$ that is identical with $a^*$ receives a score of one. This function is primarily applicable to intent types whose responses follow a more standardized format, similar to the evaluation on question answering literature (Rajpurkar et al., 2016; Yang et al., 2018).

Second, we create `must_include` where any $\hat{a}$ containing $a^*$ receives a score of one. This function is primarily used in when an unordered list of text is expected or where the emphasis of evaluation is on certain key concepts. In the second example in Table 1, we expect both the correct name and the email address to be presented, irrespective of the precise wording used to convey the answer.

Finally, we introduce `fuzzy_match` where we utilize a language model to assess whether $\hat{a}$ is semantically equivalent to $a^*$. Specifically, in this work, we use `gpt-4-0613` to perform this evaluation. The corresponding prompt details are provided in Appendix A.7. The `fuzzy_match` function applies to situations where the format of the answer is diverse. For instance, in responding to "*Compare the time for walking and driving route from AMC Waterfront to Randyland*", it is essential to ensure that driving time and walking time are accurately linked with the correct terms. The `fuzzy_match` function could also flexibly match the time "2h58min" with different forms such as "2 hour 58 minutes", "2:58" and others. We demonstrate a language model can achieve nearly perfect performance on this task in §A.8.

**Evaluating Site Navigation and Content & Config Tasks**   The tasks in these categories require accessing web pages that meet certain conditions or performing operations that modify the underlying data storage of the respective websites. To assess these, we establish reward functions $r_{\text{prog}}(\mathbf{s})$ that programmatically examine the intermediate states $\mathbf{s}$ within an execution trajectory to ascertain whether the outcome aligns with the intended result. These intermediate states are often the underlying databases of the websites, the status, and the content of a web page at each step of the execution.

Evaluating each instance involves two components. First, we provide a `locator`, tasked with retrieving the critical content pertinent to each intent. The implementation of this locator varies from a database query, a website-supported API call, to a JavaScript element selection on the relevant web page, depending on implementation feasibility. For example, the evaluation process for the intent of the fifth example in Table 1, first obtains the URL of the latest post by examining the last state in the state sequence $\mathbf{s}$. Then it navigates to the corresponding post page and obtains the post's content by running the Javascript "`document.querySelector('.submission__inner').outerText`".

Subsequently, we annotate `keywords` that need to exist within the located content. For example, the evaluation verifies if the post is correctly posted in the "nyc" subreddit by examining the URL of the post and if the post contains the requested content by examining the post content. We reuse the `exact_match` and `must_include` functions from information-seeking tasks for this purpose.

| Function | ID | Intent | Eval Implementation |
|---|---|---|---|
| $r_{\text{info}}(a^*, \hat{a})$ | 1 | Tell me the name of the customer who has the most cancellations in the history | `exact_match`($\hat{a}$, "Samantha Jones") |
| | 2 | Find the customer name and email with phone number 8015551212 | `must_include`($\hat{a}$, "Sean Miller")
`must_include`($\hat{a}$, "sean@gmail.com") |
| | 3 | Compare walking and driving time from AMC Waterfront to Randyland | `fuzzy_match`($\hat{a}$, "walking: 2h58min")
`fuzzy_match`($\hat{a}$, "driving: 21min") |
| $r_{\text{prog}}(\mathbf{s})$ | 4 | Checkout merge requests assigned to me | `url=locate_current_url`(**s**)
`exact_match`(URL, "gitlab.com/merge_requests?assignee_username=byteblaze") |
| | 5 | Post to ask "whether I need a car in NYC" | `url=locate_latest_post_url`(**s**)
`body=locate_latest_post_body`(**s**)
`must_include`(URL, "/f/nyc")
`must_include`(body, "a car in NYC") |

Table 1: We introduce two evaluation approaches. $r_{\text{info}}$ (top) measures the correctness of performing information-seeking tasks. It compares the predicted answer $\hat{a}$ with the annotated reference $a^*$ with three implementations. $r_{\text{prog}}$ (bottom) programmatically checks whether the intermediate states during the executions possess the anticipated properties specified by the intent.

**Unachievable Tasks**   Due to constraints such as inadequate evidence, user permissions (§A.3), or the absence of necessary functional support on the website, humans may ask for tasks that are not possible to complete. Inspired by previous work on evaluating question-answering models on unanswerable questions (Rajpurkar et al., 2018), we design unachievable tasks in `WebArena`. For instance, fulfilling an intent like "*Tell me the contact number of OneStopShop*" is impracticable in `WebArena`, given that the website does not provide such contact information. We label such instances as "N/A" and expect an agent to produce an equivalent response. These examples allow us to assess an agent's ability to avoid making unfounded claims and its adherence to factual accuracy.

**Annotation Process**   The intents were contributed by the authors following the annotation guideline in §3.1. Every author has extensive experience with web-based tasks. The reference answers to the information-seeking tasks were curated by the authors and an external annotator. To ensure consistency and accuracy, each question was annotated twice. If the two annotators disagreed, a third annotator finalized the annotation. The programs to evaluate the remaining examples were contributed by three of the authors who are proficient in JavaScript programming. Difficult tasks were often discussed collectively to ensure the correctness of the annotation. The annotation required the annotator to undertake the full execution and scrutinize the intermediate states.

**Human Performance**   We sample one task from each of the 170 templates and ask five computer science graduate students to perform these tasks. The human performance is on the right. Overall, the human annotators complete 78.24% of the tasks, with lower performance on information-seeking tasks. Through examining the recorded trajectories,

| | |
|---|---|
| Avg. Time | 110s |
| Success Rate$_{\text{info}}$ | 74.68% |
| Success Rate$_{\text{others}}$ | 81.32% |
| Success Rate$_{\text{all}}$ | 78.24% |

we found that 50% of the failures are due to misinterpreting the intent (*e.g.,* providing travel distance when asked for travel time), incomplete answers (*e.g.,* providing only name when asked for name and email), and incomplete executions (*e.g.,* partially filling the product information), while the remaining instances have more severe failures, where the executions are off-target. More discussions on human annotations can be found in §A.5.

## 4   BASELINE WEB AGENTS

We experiment with three LLMs using two prompting strategies, both with two examples in the context. In the first setting, we ask the LLM to directly predict the next action given the current observation, the intent and the previously performed action. In the second setting, with the same information, the model first performs chain-of-thought reasoning steps in the text before the action prediction (CoT, Wei et al. (2022); Yao et al. (2022b)). Before the examples, we provide a detailed overview of the browser environment, the allowed actions, and many rules. To make the model aware of the unachievable tasks, the instruction explicitly asks the agent to stop if it believes the task is impossible to perform. We refer to this directive as Unachievable hint, or **UA hint**. This introduction

is largely identical to the guidelines we presented to human annotators to ensure a fair comparison. We use an accessibility tree with element IDs as the observation space. The agent can identify which element to interact with by the ID of the element. For instance, the agent can issue `click [1582]` to click the "Add to Cart" button with the ID of 1582. The full prompts can be found in Appendix A.9. The detailed configurations of each model can be found in Appendix A.6.

## 5 RESULTS

The main results are shown on the top of Table 2. GPT-4 (OpenAI, 2023) with CoT prompting achieves a modest end-to-end task success rate of 11.70%, which is significantly lower than the human performance of 78.24%. GPT-3.5 (OpenAI, 2022) with CoT prompting is only able to successfully perform 8.75% of the tasks. The explicit reasoning procedure is somewhat helpful, it brings 2.34% improvement over the version without it. Further, TEXT-BISON-001 (Anil et al., 2023) underperforms GPT-3.5, with a success rate of 5.05%. These results underline the inherent challenges and complexities of executing tasks that span long horizons, particularly in realistic environments such as `WebArena`.

| CoT | UA Hint | Model | SR | SR$_{AC}$ | SR$_{UA}$ |
|---|---|---|---|---|---|
| ✓ | ✓ | TEXT-BISON-001 | 5.05 | 4.00 | 27.78 |
| ✗ | ✓ | GPT-3.5 | 6.41 | 4.90 | 38.89 |
| ✓ | ✓ | GPT-3.5 | 8.75 | 6.44 | 58.33 |
| ✓ | ✓ | GPT-4 | 11.70 | 8.63 | **77.78** |
| ✗ | ✗ | GPT-3.5 | 5.10 | 4.90 | 8.33 |
| ✓ | ✗ | GPT-3.5 | 6.16 | 6.06 | 8.33 |
| ✓ | ✗ | GPT-4 | **14.41** | **13.02** | 44.44 |
| - | ✓ | Human | 78.24 | 77.30 | 100.00 |

Table 2: The end-to-end task success rate (SR %) on `WebArena` with different prompting strategies. **CoT**: the model performs step-by-step reasoning before issuing the action. **UA hint**: ask the model to stop when encountering unachievable questions.

### 5.1 ANALYSIS

**Do models know when to stop?** In our error analysis of the execution trajectories, we observe a prevalent error pattern of early stopping due to the model's conclusion of unachievability. For instance, GPT-4 erroneously identifies 54.9% of feasible tasks as impossible. This issue primarily stems from the UA hint in the instruction, while this hint allows models to identify unachievable tasks, it also hinders performance on achievable tasks. To address this, we conduct an ablation study where we remove this hint. We then break down the success rate for both achievable and unachievable tasks. As shown in Table 2, eliminating this instruction led to a performance boost in achievable tasks, enhancing the overall task success rate of GPT-4 to 14.41%. Despite an overall decline in identifying unachievable tasks, GPT-4 retains the capacity to recognize 44.44% of such tasks. It does so by generating *reasons of non-achievability*, even without explicit instructions. On the other hand, GPT-3.5 rarely exhibits this level of reasoning. Instead, it tends to follow problematic patterns such as hallucinating incorrect answers, repeating invalid actions, or exceeding the step limits. This result suggests that even subtle differences in instruction design can significantly influence the behavior of a model in performing interactive tasks in complex environments.

**Can a model maintain consistent performance across similar tasks?** Tasks that originate from the same template usually follow similar reasoning and planning processes, even though their observations and executions will differ. We plot a histogram of per-template success rates for our models in Table 3. Of the 61 templates, GPT-4 manages to achieve a 100% task success rate on only four templates, while GPT-3.5 fails to achieve full task completion for any of the templates. In many cases, the models are only able to complete one task variation with a template. These observations indicate that even when tasks are derived from the same template, they can present distinct challenges. For instance, while "*Fork metaseq*" can be a straightforward task, "*Fork all repos from Facebook*" derived from the

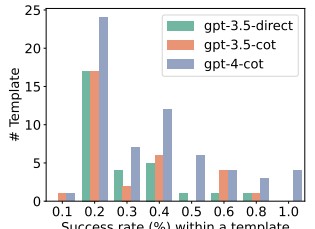

Table 3: Distribution of success rates on templates with ≥ 1 successful executions on GPT models (no UA hint).

same template requires more repetitive operations, hence increasing its complexity. Therefore, `WebArena` provide a testbed to evaluate more sophisticated methods. In particular, those that incorporate memory components,

| Benchmark | | Dynamic Interaction? | Realistic Environment? | Diverse Human Tasks? | Functional Correctness? |
|---|---|:---:|:---:|:---:|:---:|
| Mind2Web | (Deng et al., 2023) | ✗ | ✓ | ✓ | ✗ |
| Form/QAWoB | (Shi et al., 2017) | ✗ | ✓ | ✓ | ✗ |
| MiniWoB++ | (Liu et al., 2018) | ✓ | ✗ | ✗ | ✓ |
| Webshop | (Yao et al., 2022a) | ✓ | ✗ | ✗ | ✓ |
| ALFRED | (Shridhar et al., 2020) | ✓ | ✗ | ✗ | ✓ |
| VirtualHome | (Puig et al., 2018) | ✗ | ✗ | ✓ | ✗ |
| AndroidEnv | (Toyama et al., 2021) | ✓ | ✓ | ✗ | ✗ |
| WebArena | | ✓ | ✓ | ✓ | ✓ |

Table 4: The comparison between our benchmark and existing benchmarks on grounding natural language instructions to concrete executions. Our benchmark is implemented in our fully interactable highly-realistic environment. It features diverse tasks humans may encounter in their daily routines. We design evaluation metrics to assess the functional correctness of task executions.

enabling the *reuse* of successful strategies from past experiments Zhou et al. (2022a); Wang et al. (2023). More error analysis with examples can be found in Appendix A.10.

## 6 RELATED WORK

**Benchmarks for Controlling Agents through Natural Language**   Controlling agents via natural language in the digital world have been studied in the literature (Branavan et al., 2009; Shi et al., 2017; Liu et al., 2018; Toyama et al., 2021; Deng et al., 2023; Li et al., 2020; Xu et al., 2021). However, the balance between *functionality*, *authenticity*, and *support for environmental dynamics* remains a challenge. Existing benchmarks often compromise these aspects, as shown in Table 4. Some works rely on static states, limiting agents' explorations and functional correctness evaluation (Shi et al., 2017; Deng et al., 2023), while others simplify real-world complexities, restricting task variety (Yao et al., 2022a; Liu et al., 2018). While AndroidEnv (Toyama et al., 2021) replicates an Android setup, it does not guarantee the reproducibility since live Android applications are used.  (Kolve et al., 2017; Shridhar et al., 2020; Puig et al., 2018) and extends to gaming environments (Fan et al., 2022; Küttler et al., 2020), where the environment mechanisms often diverge from human objectives.

**Interactive Decision-Making Agents**   Nakano et al. (2021) introduce WebGPT which searches the web and reads the search results to answer questions. Gur et al. (2023) propose a web agent that synthesizes Javascript code for the task executions. Adding a multi-modal dimension, Lee et al. (2023) and Shaw et al. (2023) develop agents that predict actions based on screenshots of web pages rather than relying on the text-based DOM trees. Performing tasks in interactive environments requires the agents to exhibit several capabilities including hierarchical planning, state tracking, and error recovery. Existing works (Huang et al., 2022; Madaan et al., 2022; Li et al., 2023) observe LLMs could break a task into more manageable sub-tasks (Zhou et al., 2022b). This process can be further refined by representing task executions as programs, a technique that aids sub-task management and skill reuse (Zhou et al., 2022a; Liang et al., 2023; Wang et al., 2023; Gao et al., 2023). Meanwhile, search and backtracking methods introduce a more structured approach to planning while also allowing for decision reconsideration (Yao et al., 2023; Long, 2023). Existing works also incorporate failure recovery, self-correction (Shinn et al., 2023; Kim et al., 2023), observation summarization (Sridhar et al., 2023) to improve execution robustness. The complexity of WebArena presents a unique challenge and opportunity for further testing and improvement of these methods.

## 7 CONCLUSION

We present WebArena, a highly-realistic, standalone, and reproducible web environment designed for the development and testing of autonomous agents. WebArena includes fully functional web applications and organic data from popular domains. Additionally, we curate a comprehensive benchmark consisting of 812 examples that focus on mapping high-level natural language intents into concrete web interactions. We also offer outcome-based evaluation that programmatically validate the tasks success. Our experiments show that even GPT-4 only achieves a limited end-to-end task success rate of 14.41%, significantly lagging behind the human performance of 78.24%. These findings underscore the need for future research to focus on enhancing the robustness and efficacy of autonomous agents within WebArena environment.

## ACKNOWLEDGEMENT

We would like to thank Emmy Liu, Zhiruo Wang, Zhitong Guo for examining our annotations, Shunyu Yao for providing the raw Amazon product data in Webshop, Pengfei Liu, Zaid Sheikh and Aman Madaan for the helpful discussions. We are also grateful to the Center for AI Safety for providing computational resources. This material is partly based on research sponsored in part by the Air Force Research Laboratory under agreement number FA8750-19-2-0200. The U.S. Government is authorized to reproduce and distribute reprints for Governmental purposes notwithstanding any copyright notation thereon. The views and conclusions contained herein are those of the authors and should not be interpreted as necessarily representing the official policies or endorsements, either expressed or implied, of the Air Force Research Laboratory or the U.S. Government. This project was also partially supported by a gift from AWS AI.

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

# A    APPENDIX

## A.1    WEBSITE IMPLEMENTATION

Given the selected websites described in §2.2, we make the best attempt to reproduce the functionality of commonly used sites in a reproducible way. To achieve this, we utilized open-source frameworks for the development of the websites across various categories and imported data from their real-world counterparts. For the E-commerce category, we constructed a shopping website with approximately $90k$ products, including the prices, options, detailed product descriptions, images, and reviews, spanning over 300 product categories. This website is developed using Adobe Magento, an open-source e-commerce platform[4]. Data resources were obtained from data from actual online sites, such as that included in the Webshop data dumpYao et al. (2022a). As for the social forum platform, we deployed an open-source software Postmill[5], the open-sourced counterpart of Reddit[6]. We sampled from the top 50 subreddits[7]. We then manually selected many subreddit for northeast US cities as well as subreddit for machine learning and deep learning-related topics. This manual selection encourages cross-website tasks such as seeking information related to the northeast US on both Reddit and the map. In total, we have 95 subreddits, 127390 posts, and 661781 users. For the collaborative software development platform, we choose GitLab[8]. We heuristically simulate the code repository characteristics by sampling at least ten repositories for every programming language: $80\%$ of them are sampled from the set of top 90 percentile wrt stars repos using a discrete probability distribution weighted proportional to their number of stars; the remaining are sampled from the bottom ten percentile set using similar weighted distribution. This is done to ensure fair representation of repos of all kinds, from popular projects with many issues and pull requests to small personal projects. In total, we have 300 repositories and more than 1000 accounts with at least one commit to a repository. For the content management system, we adapted Adobe Magento's admin portal, deploying the sample data provided in the official guide. We employ OpenStreetMap[9] for map service implementation, confining our focus to the northeast US region due to data storage constraints. We implement a calculator and a scratchpad ourselves.

Lastly, we configure the knowledge resources as individual websites, complemented with search functionality for efficient information retrieval. Specifically, we utilize Kiwix[10] to host an offline version of English Wikipedia with a knowledge cutoff of May 2023. The user manuals for GitLab and Adobe Commerce Merchant documentation are scraped from the official websites.

## A.2    ENVIRONMENT DELIVERY AND RESET

One goal for our evaluation environment is ease of use and reproducibility. As a result, we deploy our websites in separate Docker images [11], one per website. The Docker images are fully self-contained with all the code of the website, database, as well as any other software dependencies. They also do not rely on external volume mounts to function, as the data of the websites are also part of the docker image. This way, the image is easy to distribution containing all the pre-populated websites for reproducible evaluation. End users can download our packaged Docker images and run them on their systems and re-deploy the exact websites together with the data used in our benchmarks for their local benchmarking.

Since some evaluation cases may require the agent to modify the data contained in the website, *e.g.,* creating a new user, deleting a post, etc., it is crucial to be able to easily reset the website environment to its initial state. With Docker images, the users could stop and delete the currently running containers for that website and start the container from our original image again to fully reset the environment to the initial state. Depending on the website, this process may take from a few seconds to one minute. However, not all evaluation cases would require an environment reset, as

---

[4]`https://github.com/magento/magento2`
[5]`https://postmill.xyz/`
[6]`https://www.reddit.com/`
[7]`https://redditlist.com/sfw.html`
[8]`https://gitlab.com/gitlab-org/gitlab`
[9]`https://www.openstreetmap.org/`
[10]`https://www.kiwix.org/en/`
[11]`https://www.docker.com/`

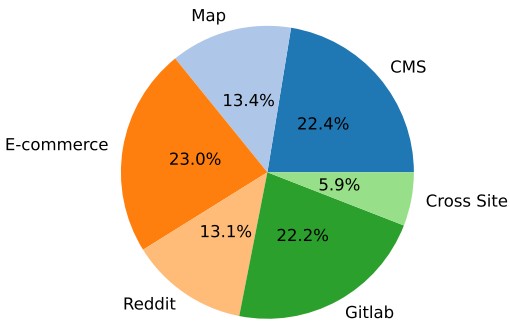

Figure 6: The intent distribution across different websites. Cross-site intents necessitate interacting with multiple websites. Notably, regardless of the website, all user intents require interactions with multiple web pages.

many of the intents are information gathering and are read-only for the website data. Also, combined with the inference time cost for the agent LLMs, we argue that this environment reset method, through restarting Docker containers from the original images, will have a non-negligible but small impact on evaluation time.

### A.3 USER ROLES SIMULATION

Users of the same website often have disparate experiences due to their distinct *roles*, *permissions*, and *interaction histories*. For instance, within an E-commerce CMS, a shop owner might possess full read and write permissions across all content, whereas an employee might only be granted write permissions for products but not for customer data. We aim to emulate this scenario by generating unique user profiles on each platform.

On the shopping site, we created a customer profile that has over 35 orders within a span of two years. On GitLab, we selected a user who maintains several popular open-source projects with numerous merge requests and issues. This user also manages a handful of personal projects privately. On Reddit, our chosen profile was a user who actively participates in discussions, with many posts and comments. Lastly, on our E-commerce CMS, we set up a user profile for a shop owner who has full read-and-write access to all system contents.

All users are automatically logged into their accounts using a pre-cached cookie. To our best knowledge, this is the first publicly available agent evaluation environment to implement such a characteristic. Existing literature typically operates under the assumption of universally identical user roles Shi et al. (2017); Liu et al. (2018); Deng et al. (2023).

### A.4 INTENT DISTRIBUTION

The distribution of intents across the websites are shown in Figure 6.

### A.5 HUMAN PERFORMANCE

We acknowledge that there may be a difference in human performance when annotators with different demographics are involved. In fact, many tasks in our dataset require domain-specific knowledge. For instance, an average user may not know what a git merge request is; or how to create a product in a complex content management system. We aim to design tasks that have easy-to-imagine outcomes (*e.g.,* a new product page is created) rather than those that are easily performed by an average user without significant domain knowledge.

| CoT | UA Hint | Model | SR |
|-----|---------|-------|-----|
| ✓ | ✗ | GPT-3.5 | 6.28 |

Table 5: The task success rate (SR %) of GPT-3.5-TURBO-16K-0613 with temperature 0.0.

| Dataset | gpt-4-0613 | gpt-4-1106-preview |
|---------|------------|--------------------|
| Date (900 examples) | 100 | 100 |
| Time duration (900 examples) | 100 | 100 |

Table 6: The accuracy (%) of two versions of GPT-4 on judging if dates and time duration of different formats are equivalent.

### A.6 EXPERIMENT CONFIGURATIONS

We experiment with GPT-3.5-TURBO-16K-0613, GPT-4-0613, and TEXT-BISON-001 with a temperature of 1.0 and a top-$p$ parameter of 0.9. The maximum number of state transitions is set to 30. We halt execution if the same action is repeated more than three times on the same observation or if the agent generates three consecutive invalid actions. These situations typically indicate a high likelihood of execution failure and hence warrant early termination. For TEXT-BISON-001, we additionally allow ten retries until it generates a valid action.

Primarily, we use a high temperature of 1.0 to encourage the *exploration*. To aid replicating the results, we provide the results of GPT-3.5-TURBO-16K-0613 with temperature 0.0 in Table 5 and the execution trajectories in our code repository.

### A.7 PROMPT FOR FUZZY_MATCH

> Help a teacher to grade the answer of a student given a question. Keep in mind that the student may use different phrasing or wording to answer the question. The goal is to evaluate whether the answer is semantically equivalent to the reference answer.
> question: {{intent}}
> reference answer: {{reference answer}}
> all the string 'N/A' that you see is a special sequence that means 'not achievable'
> student answer: {{prediction}}
> Conclude the judgement by correct/incorrect/partially correct.

Predictions that are judged as "correct" will receive a score of one, while all other predictions will receive a score of zero.

### A.8 THE ACCURACY OF FUZZY MATCH FUNCTION

To evaluate this, we manually checked 40 examples and found that 39 of them are identical to our human judgment. In addition, among the 82 examples that require using GPT-4 for evaluation, the answer of 49 (60%) examples is a date (*e.g.,* 10/23/2022) or time duration (*e.g.,* 15 minutes). In these cases, GPT-4 is only used to judge the different *format* of the answers. We quantitatively evaluate the correctness of GPT-4 in this case by generating different formats of a date and time duration programmatically. We randomly sample negative examples. For instance, Nov 3, 2022, November 3, 2022, 3rd November 2022, 3 Nov 2022, 2022-11-03, and 3rd of November, 2022 are all correct variances of 2022/11/03. The accuracy of GPT-4 is shown in Table 6. We can see that two versions of GPT-4 are extremely accurate, both achieving 100% accuracy.

### A.9 THE PROMPTS OF THE BASELINE WEB AGENTS

The system message of the reasoning agent for both GPT-3.5 and GPT-4 is in Figure 7, and two examples are in Figure 8. The system message of the direct agent for GPT-3.5 is in Figure 9 and the two examples are in Figure 10. **UA hint** refers to the instruction of " If you believe the task is

You are an autonomous intelligent agent tasked with navigating a web browser. You will be given web-based tasks. These tasks will be accomplished through the use of specific actions you can issue.

Here's the information you'll have:
The user's objective: This is the task you're trying to complete.
The current web page's accessibility tree: This is a simplified representation of the webpage, providing key information.
The current web page's URL: This is the page you're currently navigating.
The open tabs: These are the tabs you have open.
The previous action: This is the action you just performed. It may be helpful to track your progress.

The actions you can perform fall into several categories:
Page Operation Actions
`click [id]`: This action clicks on an element with a specific id on the webpage.
`type [id] [content] [press_enter_after=0|1]`: Use this to type the content into the field with id. By default, the "Enter" key is pressed after typing unless press_enter_after is set to 0.
`hover [id]`: Hover over an element with id.
`press [key_comb]`: Simulates the pressing of a key combination on the keyboard (e.g., Ctrl+v).
`scroll [direction=down|up]`: Scroll the page up or down.

Tab Management Actions:
`new_tab`: Open a new, empty browser tab.
`tab_focus [tab_index]`: Switch the browser's focus to a specific tab using its index.
`close_tab`: Close the currently active tab.

URL Navigation Actions:
`goto [url]`: Navigate to a specific URL.
`go_back`: Navigate to the previously viewed page.
`go_forward`: Navigate to the next page (if a previous
`go_back` action was performed).

Completion Action:
`stop [answer]`: Issue this action when you believe the task is complete. If the objective is to find a text-based answer, provide the answer in the bracket. If you believe the task is impossible to complete, provide the answer as "N/A" in the bracket.

Homepage:
If you want to visit other websites, check out the homepage at http://homepage.com. It has a list of websites you can visit.
http://homepage.com/password.html lists all the account names and passwords for the websites. You can use them to log in to the websites.

To be successful, it is very important to follow the following rules:
1. You should only issue an action that is valid given the current observation
2. You should only issue one action at a time.
3. You should follow the examples to reason step by step and then issue the next action.
4. Generate the action in the correct format. Start with a "In summary, the next action I will perform is" phrase, followed by action inside ``````. For example, "In summary, the next action I will perform is ```click [1234]```".
5. Issue stop action when you think you have achieved the objective. Don't generate anything after stop."""

Figure 7: The `system` message of the reasoning agent. This message has a general explanation of the task, the available actions, and some notes on avoiding common failures.

impossible to complete, provide the answer as "N/A" in the bracket.". We remove this sentence in our ablation studies.

example_user
OBSERVATION:
[1744] link 'HP CB782A#ABA 640 Inkjet Fax Machine (Renewed)'
  [1749] StaticText '$279.49'
  [1757] button 'Add to Cart'
  [1760] button 'Add to Wish List'
  [1761] button 'Add to Compare'
URL: http://onestopmarket.com/office-products/office-electronics.html
OBJECTIVE: What is the price of HP Inkjet Fax Machine
PREVIOUS ACTION: None

example_assistant
Let's think step-by-step. This page lists the information of HP Inkjet Fax Machine, which is the product identified in the objective. Its price is $279.49. I think I have achieved the objective. I will issue the stop action with the answer. In summary, the next action I will perform is ```stop [$279.49]```

example_user
OBSERVATION:
[164] textbox 'Search' focused: True required: False
[171] button 'Go'
[174] link 'Find directions between two points'
[212] heading 'Search Results'
[216] button 'Close'
URL: http://openstreetmap.org
OBJECTIVE: Show me the restaurants near ABC
PREVIOUS ACTION: None

example_assistant
Let's think step-by-step. This page has a search box whose ID is [164]. According to the nominatim rule of openstreetmap, I can search for the restaurants near a location by "restaurants near." I can submit my typing by pressing the Enter afterwards. In summary, the next action I will perform is ```type [164] [restaurants near ABC] [1]```

Figure 8: The two examples provided as `example_user` and `example_assistant` for the reasoning agent. Before issuing the action, the agent first perform reasoning.

You are an autonomous intelligent agent tasked with navigating a web browser. You will be given web-based tasks. These tasks will be accomplished through the use of specific actions you can issue.

Here's the information you'll have:
The user's objective: This is the task you're trying to complete.
The current web page's accessibility tree: This is a simplified representation of the webpage, providing key information.
The current web page's URL: This is the page you're currently navigating.
The open tabs: These are the tabs you have open.
The previous action: This is the action you just performed. It may be helpful to track your progress.

The actions you can perform fall into several categories:
Page Operation Actions
`click [id]`: This action clicks on an element with a specific id on the webpage.
`type [id] [content] [press_enter_after=0|1]`: Use this to type the content into the field with id. By default, the "Enter" key is pressed after typing unless press_enter_after is set to 0.
`hover [id]`: Hover over an element with id.
`press [key_comb]`: Simulates the pressing of a key combination on the keyboard (e.g., Ctrl+v).
`scroll [direction=down|up]`: Scroll the page up or down.

Tab Management Actions:
`new_tab`: Open a new, empty browser tab.
`tab_focus [tab_index]`: Switch the browser's focus to a specific tab using its index.
`close_tab`: Close the currently active tab.

URL Navigation Actions:
`goto [url]`: Navigate to a specific URL.
`go_back`: Navigate to the previously viewed page.
`go_forward`: Navigate to the next page (if a previous
`go_back` action was performed).

Completion Action:
`stop [answer]`: Issue this action when you believe the task is complete. If the objective is to find a text-based answer, provide the answer in the bracket. If you believe the task is impossible to complete, provide the answer as "N/A" in the bracket.

Homepage:
If you want to visit other websites, check out the homepage at http://homepage.com. It has a list of websites you can visit.
http://homepage.com/password.html lists all the account name and password for the websites. You can use them to log in to the websites.

To be successful, it is very important to follow the following rules:
To be successful, it is very important to follow the following rules:
1. You should only issue an action that is valid given the current observation
2. You should only issue one action at a time.
3. Generate the action in the correct format. Always put the action inside a pair of ` ` `. For example, ` ` `click [1234]` ` `
4. Issue stop action when you think you have achieved the objective. Don't generate anything after stop."""

Figure 9: The `system` message of the direct agent. This message has the general explanation of the task, the available actions and some notes on avoiding common failures.

example_user
OBSERVATION:
[1744] link 'HP CB782A#ABA 640 Inkjet Fax Machine (Renewed)'
  [1749] StaticText '$279.49'
  [1757] button 'Add to Cart'
  [1760] button 'Add to Wish List'
  [1761] button 'Add to Compare'
URL: http://onestopmarket.com/office-products/office-electronics.html
OBJECTIVE: What is the price of HP Inkjet Fax Machine
PREVIOUS ACTION: None

example_assistant
```stop [$279.49]```

example_user
OBSERVATION:
[164] textbox 'Search' focused: True required: False
[171] button 'Go'
[174] link 'Find directions between two points'
[212] heading 'Search Results'
[216] button 'Close'
URL: http://openstreetmap.org
OBJECTIVE: Show me the restaurants near ABC
PREVIOUS ACTION: None

example_assistant
```type [164] [restaurants near ABC] [1]```

Figure 10: The two examples provided as `example_user` and `example_assistant` for the direct agent. The agent directly emits the next action given the observation.

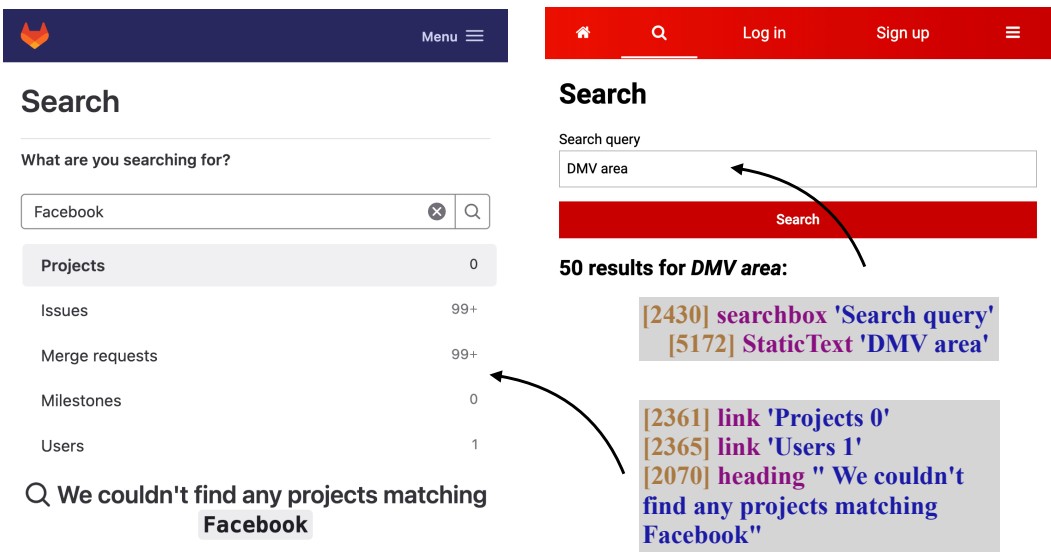

Figure 11: Two examples where the GPT-4 agent failed, along with their screenshot and the accessibility tree of the relevant sections (grey). On the left, the agent fails to proceed to the "Users" section to accomplish the task of "Fork all Facebook repos"; on the right, the agent repeats entering the same search query even though the observation indicates the input box is filled.

## A.10 ADDITIONAL ERROR ANALYSIS

**Observation Bias** Realistic websites frequently present information on similar topics across various sections to ensure optimal user accessibility. However, a GPT-4 agent often demonstrates a tendency to latch onto the first related piece of information it encounters without sufficiently verifying its relevance or accuracy. For instance, the homepage of the E-Commerce CMS displays the best-selling items based on *recent purchases*, while historical best-seller data is typically accessed via a separate report. Presented with the task of "*What is the top-1 best-selling product in 2022*", the GPT-4 agent defaults to leveraging the readily available information on the homepage, bypassing the necessary step of generating the report to obtain the accurate data.

**Failures in Observation Interpretation** Interestingly, while GPT-4 is capable of summarizing the observations, it occasionally overlooks more granular information, such as the previously entered input. As in the right-hand example of Figure 11, `[5172] StaticText` indicates that the search term "DMV area" has already been entered. However, the agent disregards this detail and continuously issues the command `type [2430] [DMV area]` until it reaches the maximum step limit. Furthermore, the agent often neglects the previous action information that is provided alongside the observation.

We hypothesize that these observed failures are related to the current pretraining and supervised fine-tuning on dialogues employed in GPT models Ouyang et al. (2022). These models are primarily trained to execute instructions given *immediate* observations (*i.e.,*, the dialogue history); thereby, they may exhibit a lack of explorations. Furthermore, in dialogue scenarios, subtle differences in NL expressions often have less impact on the overall conversation. As a result, models may tend to overlook minor variations in their observations.