# OpenReview forum: "WebArena: A Realistic Web Environment for Building Autonomous Agents"
_ICLR.cc/2024/Conference — ICLR 2024 poster_

### Official Review · Reviewer_Y9es · 2023-10-30

**Soundness:** 2 fair
**Presentation:** 3 good
**Contribution:** 2 fair
**Rating:** 5
**Confidence:** 3

**Summary:**

This paper proposes a web environment designed for the development and testing of autonomous agents. The proposed environment WebArena includes fully functional web applications and genuine data from four major categories, providing a realistic platform for agent interaction. The authors also proposes a benchmark consisting of 812 examples, as well as an evaluation method. The experiments show that GPT-4 only achieves a task success rate of 14.41%, which is much lower than human performance of 78.24%.

**Strengths:**

1. This paper proposes a highly-realistic and complicated web environment compared with the previous simplified environment;

2. The proposed environment includes four common and real domains;

3. The paper is well written and easy to follow.

**Weaknesses:**

1. The major weakness of this paper is the lack of technical novelty. Though the contribution on simulated environment/datasets/resources are welcomed and very important to the research community, such papers may not match the general style of ICLR papers.

2. For evaluation, the proposed framework uses GPT4 to evaluate the answer or the execution paths, which potentially has two issues: 1. GPT4 is a commercial tool, which may limit the potential use of this environment; 2. GPT4 is not guaranteed to be 100% right, which may make the evaluation results not convincing.

3. The success rate of human on the designed tasks are only 78%, which is a little surprising since it seems that these tasks are not that difficult for human to complete. It is better that the authors provide more analysis on these tasks and evaluations to show that why human fails and if these tasks are too difficult for agents.

**Questions:**

1. Is there any analysis or discussion on the performance of GPT4 evaluation?

2. Why the success rate of human is only 78%?

---

> ### Author Response · Authors · 2023-11-21
> **Response to Reviewer Y9es**
>
> Thank you for taking the time to review our paper and for your helpful feedback. We are happy that you appreciated the realism and the complexity of the benchmark!
>
> We think that all your questions are addressable within this discussion period. Please see our response below. We will be happy to address additional questions during the discussion period if anything is unclear.
>
> > ## Environment/datasets/resources do not fit the style of ICLR
>
> We kindly note the primary area within ICLR that this paper is submitted to is **datasets and benchmarks**, and we believe our work is a good fit for it. Please see  the ICLR 2024 call for papers (https://iclr.cc/Conferences/2024/CallForPapers) where “datasets and benchmarks” is listed as one of the 20 topics of interest.
>
> > ## Will using GPT-4 for evaluation limit the use of the WebArena environment?
>
> Thanks for pointing this out. We argue that using GPT-4 will *not* prevent future research on WebArena. Only ~10% (82 out of 812) of the examples require using GPT-4 for evaluation, and the cost is less than 1 US dollar.
>
> > ## Is GPT-4 evaluation accurate?
>
> This is a reasonable concern. To evaluate this, we manually checked 40 examples and found that 39 of them are identical to our human judgment. In addition, among the 82 examples that require using GPT-4 for evaluation, the answer of 49 (60%) examples is a date (e.g., 10/23/2022) or time duration (e.g., 15 minutes). In these cases, GPT-4 is only used to judge the different **format** of the answers. We quantitatively evaluate the correctness of GPT-4 in this case by generating different formats of a date and time duration programmatically. We randomly sample negative examples. For instance, `Nov 3, 2022`, `November 3, 2022`, `3rd November 2022`, `3 Nov 2022`, `2022-11-03`, and `3rd of November, 2022` are all correct variances of `2022/11/03`. The accuracy of GPT-4 is as follows:
>
> |                         | gpt-4-0613 (same as in the paper) | gpt-4-1106-preview (GPT-4 Turbo) |
> |-------------------------|-----------------------------------|----------------------------------|
> | Date (900 examples)     | 100%                              | 100%                             |
> | Time duration (900 examples) | 100%                              | 100%                             |
>
>
> We can see that GPT-4 and its recent turbo version are extremely accurate, both achieving 100% accuracy. We will add these experiments to our revision.
>
>
> > ## Where do humans fail?
>
> As described in our paper (Section 3.2), we found that 50% of the failures are not due to human annotators’ inability to perform the tasks. The major errors are (1) misinterpreting the intent such as providing the travel distance when asked for travel time. (2) Incomplete answers such as providing only the customer's name when asked for name and email. And (3) incomplete executions such as partially filling the product information. The other 50% are more severe failures where the human performed the task in a wrong way.

---

### Official Review · Reviewer_uu7U · 2023-11-01

**Soundness:** 2 fair
**Presentation:** 3 good
**Contribution:** 3 good
**Rating:** 6
**Confidence:** 3

**Summary:**

The paper proposes a new, realistic RL envionrment for Web tasks named WebArena as well as a first evaluation of GPT-based agents performing the defined tasks. The framework includes environments for e-commerce, social forums, collaborative software development similar to Gitlab and content management and therefore provides additional tools, including maps or Wikis. The obervation space can be screenshots of web pages, HTML DOM trees or accessibility trees. The authors proposes a Partially Observable Markov Decision Process modelling of tasks, where the action space comprises keyboard and mouse. The authors present 812 tasks in their benchmark and evaluate GPT-based agents for the tasks, which yield sub-par performance against human baselines from a user study.

**Strengths:**

The authors propose an Independent platform, implementing a large variety of realistic end-user tasks on the Web. The framework provides provides realistical, challenging tasks for Web agents. The quality of the benchmark is sufficiently high. To this end, a good choice of task variety was made, which is backed up by a user study. This is very nice to see, as the taken design decisions then are probabily matching with user needs.

The paper includes a preliminary evaluation of agents based on closed-source LLMs (ChatGPT / Text-Bison), which gives first insights.

The code is made available for review, and is usable and documented, which makes the paper's contributions quite clear. The available tasks are sufficiently challenging for evaluating (LLM) agents, which makes the contributions significant for more research advances in the field.

While there are other related benchmarks in the field, it is quite clear from the paper content what is being improved / what is original.

**Weaknesses:**

The related work advantage not completely clear. The related work states functional correctness as advantage over AndroidEnv, but no further explanation is given. It might hint to the diffeence between the used evaluation metrics, but it would be interesting/important to clarify this. Also, it mentions the lack of diverse or complex task availability, but new tasks can be defined within the framework.

The agent evaluation is performaned with standard GPT variants only, not pointing to stronger alternatives. Also, little to no details about how the agent was implemented/tested are given in the main paper. Only the appendix shows examples, which impedes understanding the paper.

As the used LLMs for the evaluation are closed-source, this impedes reproduciability. As the evaluation can been seen as first validation of the benchmark, this might still be fine, but it would be good to have open-source agents integrated.

Lastly, the POMDP model is not argued for in the paper, but it would be important to justify the modelling choice. This is not to say that a POMDP model is not sensible.

**Questions:**

Can new tasks be easily added to the benchmark within the available environments?

Would it it have been possible to include the benchmark tasks into another, existing benchmark system from the related work?

What could be future works wrt to (RL-) agents for solving the benchmark tasks?

Are the presented tasks on par wrt difficulty or even superior to other benchmarks?

---

> ### Author Response · Authors · 2023-11-21
> **Response to Reviewer uu7U (1/2)**
>
> Thank you for taking the time to review our paper, for your constructive feedback, and for your kind words regarding the high quality of the benchmark!
>
> We think that all your questions are addressable within this discussion period. Please see our response below. We will be happy to address additional questions during the discussion period if anything is unclear.
>
> > ## Comparison with AndroidEnv
>
> The key difference between AndroidEnv and WebArena is the **reproducibility** of the environment and tasks. The Android Apps inside AndroidEnv are live, meaning that they change over time rather than being “frozen”. Therefore, AndroidEnv faces the same technical challenges as a benchmark that was created based on live websites. For instance, CAPTCHAs, unexpected content modifications, and configuration changes. These obstacles can make it difficult to perform fair comparisons across various systems over time. As a result, creating and evaluating realistic tasks is challenging. In contrast, all these challenges are resolved in WebArena. We have updated the description of AndroidEnv in the related work section (revision in blue).
>
> Additionally, AndroidEnv focuses on Android applications, while WebArena focuses on browser-based applications. While there are some similarities in the tasks, there are also differences, such as UI layouts, that make the two settings unique.
>
>
> > ## How do open-source models perform on WebArena?
>
> We are delighted that many follow-up works (not by us) have evaluated open-source models on WebArena after our public release. Because we are not sure whether directly citing the works will break the anonymity policy, we only report the numbers here. We will cite these works in the camera-ready version.
>
> | Model                  | Task Success Rate (%) |
> |------------------------|-----------------------|
> | CodeLLama-34B-Instruct | 4.06                  |
> | LLama-2-Chat-7B        | 1.23                  |
> | LLama-2-Chat-34B       | 1.11                  |
> | LLama-2-Chat-70B       | 1.72                  |
>
>
> Overall, WebArena presents a significant challenge for both closed and open models.
>
> > ## Do you evaluate agents other than CoT and Direct?
>
> Yes, we performed preliminary experiments with agents with other prompting methods that perform planning and critique, similar to RCI [1] and Reflexion [2].  However, we did not proceed with the full experiment as the performance on a subset of examples was significantly lower.
>
> Since the tasks in WebArena span multiple pages, planning and critiquing based on the *current* observation is usually inaccurate. A general-purpose LLM (like GPT-4) does not have enough knowledge about the *specific* websites included in the WebArena benchmark. For instance, given the task of “Checkout my pending orders”, the agent first plans to perform the action `click [1283] to go to the "My Account" page.`, its critique component responds that `It assumes that clicking on the "My Account" link ([1283]) will take you to the page where pending orders are displayed. However, there is no guarantee that this is the case. The "My Account" page could have a different purpose or may not even exist.`. However, there is indeed a “My order” link inside “My Account”. Furthermore, the plan is usually vague and it does not give enough hints for the agent to perform the concrete next step. We found that the agent repeated the same action more frequently and failed to execute the given task.
>
> We will add this discussion to the Appendix in our revision.
>
>
> > ## Implementation and evaluation details of the agents
>
> As described at the beginning of Section 4, we performed evaluations under a few-shot in-context learning setting. We did not perform any finetuning which updates the parameters. The prompts with or without CoT are listed in Appendix A.5, but we agree that it may make the paper clearer if we move additional details to the main paper. We will consider this for the final version, depending on how much space we can make in the paper.
>
> The evaluation process is described in Section 3.2 and concrete examples in Table 1.
>
> > ## The formulation of POMDP
>
> We originally used the POMDP formulation because it is a standard way of modeling these problems in related works (e.g., WebShop [2]). However, we agree that this formulation can be confusing. Hence, we decided to remove the mention of POMDP and rephrase the task-solving process in a more simple way as follows:
> > Given a task described as a natural language intent $\mathbf{i}$, an agent issues an action $a_t \in \mathcal{A}$ based on intent $\mathbf{i}$, the current observation $o_t \in \mathcal{O}$, the action history and the observation history $a_{1}^{t-1}, o_{1}^{t-1}$
>
> Section 2.1 has been updated accordingly.

---

> > ### Author Response · Authors · 2023-11-21
> > **Response to Reviewer uu7U (2/2)**
> >
> > > ## Can new tasks be easily added to the benchmark within the available environments?
> >
> > Yes. We can easily add new tasks following the process described in Sec.3.1 and Sec.3.2 of the paper. There is no need to modify anything in the existing environment.  For instance, an information-seeking task only requires the question and the corresponding annotated answer.
> >
> > > ## Would it have been possible to include the benchmark tasks into another, existing benchmark system from the related work?
> >
> > Yes. The observation space and action space in WebArena are generic. Adding other web-based benchmarks, such as Mind2Web is a matter of engineering.
> >
> > > ## Can you apply RL-agent on WebArena?
> >
> > RL definitely has a lot of potential in solving tasks in WebArena. For instance, can we leverage the prior knowledge encoded in LLMs to shape a better reward model? Can we have more efficient exploration in WebArena’s huge state space? However, the main point of our paper is focused on building an extensive benchmark and using it to evaluate state-of-the-art LLMs (such as GPT-4), so we hope to leave methodological improvements such as this to future work.
> >
> > > ## How do the difficulty levels of tasks in WebArena compare to those in other existing benchmarks?
> >
> > Tasks in WebArena are more challenging than tasks in simplified synthetic environments. For instance, GPT-4 can achieve more than 95% success rate on MiniWob++[1], while it only achieves a success rate of ~15% on WebArena. Our task difficulty is on par or more challenging than Mind2Web as we require end-to-end task success instead of checking if an agent produces the correct action strings.
> >
> > [1] Synapse: Trajectory-As-Exemplar Prompting With Memory For Computer Control, Zheng el at， 2023
> >
> > [2] WebShop: Towards Scalable Real-World Web Interaction with Grounded Language Agents, Yao el at, 2022

---

### Official Review · Reviewer_JxdP · 2023-11-01

**Soundness:** 3 good
**Presentation:** 4 excellent
**Contribution:** 4 excellent
**Rating:** 8
**Confidence:** 4

**Summary:**

The paper emphasizes the potential of generative AI in creating autonomous agents that can handle daily tasks using natural language commands. Recognizing the limitations of current synthetic environments, the authors introduce "WebArena," a realistic and reproducible web environment. This environment hosts websites from four key domains: e-commerce, social forums, collaborative software development, and content management, and is equipped with tools and knowledge bases to support human-like task performance. The authors also provide a benchmark of diverse tasks that mimic human internet activities and prioritize evaluating functional correctness over mere textual similarity. Testing with agents, including a GPT-4-based one, showed a significant performance gap, with the agent achieving only a 14.41% success rate compared to humans at 78.24%. This underscores the need for enhanced agent development and the value of WebArena as a testing ground for future advancements.

**Strengths:**

**Originality**: I am truly delighted to have the opportunity to review this work. I had the privilege of reading this manuscript a few months ago, and its significance resonated with me. The issues addressed in this paper are both critical and captivating. The work notably bridges a substantial gap, laying a pivotal foundation for future industrial applications of web agents. Over the last six months, I've come across numerous works on agent benchmarks. However, this particular study stands out, primarily due to its compelling motivation and remarkable originality. It has quickly become one of my favored works in this domain.

**Quality**: After personally setting up the environment, running the provided code, and assessing the dataset, I can attest to the high caliber of this work. The construction of the benchmark is solid and robust, testifying to the meticulous efforts behind it.

**Clarity**: The paper is lucidly crafted with a coherent structure and logical flow, making it accessible and comprehensible.

In conclusion, this is a high-quality, original, clear, and significantly impactful piece of scholarship.

**Weaknesses:**

While the work presented is undeniably valuable, from an academic perspective, I believe there are several weaknesses, primarily related to experimental evaluations and the choice of baselines. Here are the specific areas of concern:

1. **Lack of Evaluation with the Latest Intelligent Agents**:

The paper seems to miss out on evaluating some of the latest intelligent agents, especially those grounded in modern reasoning and planning methods. Works like the "Tree of Thought" and the new "Reflection" architecture have been in the public domain for a while. It would have greatly enhanced the paper's comprehensiveness if these contemporary agents were included in the evaluations.

2. **API Call Methodology and Ablation Experiments**:

The manner in which API calls are presented in the paper, particularly as web pages, does not seem to align with the current prevalent paradigms where APIs are usually invoked within context. It raises the question of whether an agent can effectively utilize this format. Additionally, it would have been illuminating if the authors had included ablation studies in their experiments. Specifically, it would be insightful to discern the efficacy of these tools and whether they genuinely aid the agent in realizing the desired goal of "encouraging human-like task-solving".

3. **Html or Accessibility Tree?**
   - Many language models (LLMs) are pre-trained with an abundance of HTML content, but they might not necessarily contain the Accessibility tree. Hence, it might be more natural for these LLMs to understand and parse HTML.
   - Both DOM and the Accessibility tree adopt a tree-like structure. The seemingly "redundant" symbols in HTML could potentially assist LLMs in better understanding the hierarchical nature of the content.
   - It is vital to conduct empirical tests to validate the advantages of either approach. Given that the Accessibility tree is not commonly adopted in other benchmarks, using it here could also be viewed as one of the paper's core contributions, setting it apart from the current landscape of research in this area.

4. **Gold Trajectories**:

The paper would benefit significantly from the inclusion of "Gold" trajectories. These trajectories can offer a benchmark for the best possible action sequences, making them an invaluable asset for future research in this domain. The absence of these trajectories is a noticeable gap in the paper.

5. **Evaluator Demographics**: The choice of computer science graduate students as evaluators raises certain concerns regarding the generalizability of the results.
   - Computer science graduate students typically possess an advanced understanding of web page interactions, which might not be representative of the average user. Their performance might be notably better than what we'd observe with a more diverse group, especially when considering common tasks like online shopping that even non-technical users frequently engage in.
   - Furthermore, it's essential to address potential biases that might arise if any of the evaluators were involved in the dataset's creation. This could compromise the validity of the scores. From a personal standpoint, I, along with several colleagues, have engaged in case studies with this dataset. Interestingly, our accuracy rates didn't match the high scores reported in the paper, which adds a touch of humor to this serious concern.

To ensure the paper's robustness and generalizability, it's crucial to address these points, preferably with empirical evidence and further discussions.

**Questions:**

See the weaknesses above.

---

> ### Author Response · Authors · 2023-11-21
> **Thank you for your postive feedback**
>
> Thank you very much for your time to review, your constructive comments, and the encouraging words!  We believe that your questions, and especially the questions regarding experiments, are addressable within this discussion period, please see our response below.
>
> > ## Do you evaluate agents other than CoT and Direct?
>
> Yes, we performed preliminary experiments with agents with other prompting methods that perform planning and critique, similar to RCI [1] and Reflexion [2].  However, we did not proceed with the full experiment as the performance on a subset of examples was significantly lower.
>
> Since the tasks in WebArena span multiple pages, planning and critiquing based on the *current* observation is usually inaccurate. A general-purpose LLM (like GPT-4) does not have enough knowledge about the *specific* websites included in the WebArena benchmark. For instance, given the task of “Checkout my pending orders”, the agent first plans to perform the action `click [1283] to go to the "My Account" page.`, its critique component responds that `It assumes that clicking on the "My Account" link ([1283]) will take you to the page where pending orders are displayed. However, there is no guarantee that this is the case. The "My Account" page could have a different purpose or may not even exist.`. However, there is indeed a “My order” link inside “My Account”. Furthermore, the plan is usually vague and it does not give enough hints for the agent to perform the concrete next step. We found that the agent repeated the same action more frequently and failed to execute the given task.
>
> We will add this discussion to the Appendix in our revision.
>
> > ## How do accessibility trees encode structures without something similar to the tags in HTML?
>
> The accessibility tree shares a similar structure to the YAML format, where indentations reflect the hierarchical structure.
>
> > ## Comparison between HTML observation and accessibility tree observation
>
> We experimented with using HTML as the observation with `gpt-3.5-turbo-16k-0613` and the results are as follows:
> |                    | Success Rate (%) |
> |----|----|
> | HTML               | 4.43             |
> | Accessibility tree | 8.75             |
>
> Empirically, we found that using accessibility trees yields better performance. Qualitatively, we found HTML is longer and contains a lot of redundancy. For instance, a single button can be nested inside several layers of HTML elements. This redundancy increases the burden on LLMs to localize and use the relevant information.
>
> We will add the results and the discussion to the Appendix in our revision.
>
> > ## Do you have golden traces?
>
> We completely agree that reference trajectories from humans could add value to the existing resources. We did make an initial attempt to provide such resources, but eventually we decided that it was not feasible within the scope of this paper due to the engineering effort involved in implementing a robust trajectory recording tool. We leave implementing such a tool and the collection of reference trajectories as our future work.
>
> > ## Is the success rate of CS graduate students higher than other evaluators?
>
> We acknowledge that there may be a difference in human performance when annotators with different demographics are involved. In fact, many tasks in our dataset require domain-specific knowledge. For instance, an average user may not know what a git merge request is; or how to create a product in a complex content management system. We aim to design tasks that have *easy-to-imagine outcomes* (e.g., a new product page is created) rather than those that are easily performed by an average user without significant domain knowledge. We will add this discussion to Section 3.2 if space permits, or to the Appendix otherwise.
>
> In addition, we attempted to resolve the potential ambiguities and ensure the annotation correctness with the process we described in Section 3.2 (Annotation process). We would love to look into the concrete examples where you and your colleagues found it difficult to perform the tasks if you don’t mind sharing the `task_id`.
>
> > ## Does the model use the tool websites as humans?
>
> In this work, our primary goal is to provide the setup of having individual tool websites for future studies. In our prompts, we simply inform the model that the homepage lists other websites rather than more explicit instructions on using some sites as tools. In this case, GPT-4 did not use these tool websites effectively. For instance, it did not use the calculator when doing tasks that require arithmetic operations. We leave more explicit prompting as an interesting and important future work.
>
> [1] Language Models can Solve Computer Tasks, Kim el at, 2023
> [2] Reflexion: Language Agents with Verbal Reinforcement Learning, Shinn el at, 2023

---

### Meta-Review · Area_Chair_3b83 · 2023-12-07

**Metareview:**

This research aims to create an environment for assessing agents' ability of performing tasks on the web. The author responses have effectively tackled the majority of the raised concerns. However, following the rebuttal phase, the remaining issues include:

1. Limited Exploration of Advantages: The paper does not extensively delve into the advantages of its proposed web-based environment compared to previous environments developed in the field. This omission hinders the perceived novelty of this work. One valid advantage highlighted by the authors is the improved reproducibility, but further discussion is needed to elucidate how this environment surpasses existing alternatives.

2. Absence of Model/Agent Comparisons: Another shortcoming is the absence of evaluation on the models or agents used in prior web-based environment research. A comparative analysis would have provided valuable insights about the challenges and values of the proposed environment.

Despite the aforementioned weaknesses, we believe that the proposed environment holds significant value for our community, and we are delighted to see this paper accepted.

**Justification For Why Not Higher Score:**

1. Limited Exploration of Advantages: The paper does not extensively delve into the advantages of its proposed web-based environment compared to previous environments developed in the field. This omission hinders the perceived novelty of this work. One valid advantage highlighted by the authors is the reproducibility, but further discussion is needed to elucidate how this environment surpasses existing alternatives.

2. Absence of Model/Agent Comparisons: Another shortcoming is the absence of evaluation on the models or agents used in prior web-based environment research. A comparative analysis would have provided valuable insights about the challenges and values of the proposed environment.

**Justification For Why Not Lower Score:**

This research aims to create an environment for assessing agents' ability of performing tasks on the web. All of us believe that the environment is useful, despite the remaining weaknesses.

The author responses have effectively tackled the majority of the raised concerns.

---

### Decision · Program_Chairs · 2024-01-16

Accept (poster)